# The Influence of the Germline *HSD3B1* Adrenal-Permissive Allele (c.1100 C) on the Somatic Alteration Landscape, the Transcriptome, and Immune Cell Infiltration in Prostate Cancer

**DOI:** 10.3390/cancers17081270

**Published:** 2025-04-09

**Authors:** Samuel Kellen, Allison Makovec, Carly D. Miller, Shayan S. Nazari, Andrew Elliott, Aiden Deacon, Emily John, Nikitha Vobugari, Neeraj Agarwal, Rana R. McKay, Pedro C. Barata, Charles J. Ryan, Nima Sharifi, Justin Hwang, Emmanuel S. Antonarakis

**Affiliations:** 1Masonic Cancer Center, University of Minnesota, Minneapolis, MN 55455, USA; kell2658@umn.edu (S.K.); makov016@umn.edu (A.M.); adeacon@umn.edu (A.D.); joh21009@umn.edu (E.J.); 2Department of Medicine, University of Minnesota-Twin Cities, Minneapolis, MN 55455, USA; vobug002@umn.edu; 3Caris Life Sciences, Phoenix, AZ 85040, USA; snazari@carisls.com (S.S.N.);; 4Division of Hematology and Oncology, Huntsman Cancer Institute, University of Utah, Salt Lake, UT 84112, USA; neeraj.agarwal@hci.utah.edu; 5Department of Radiation Medicine and Applied Sciences, UC San Diego School of Medicine, La Jolla, CA 92093, USA; rmckay@health.ucsd.edu; 6University Hospitals Seidman Cancer Center, Case Western Reserve University, Cleveland, OH 44106, USA; pedro.barata@uhhospitals.org; 7Memorial Sloan Kettering Cancer Center, New York, NY 10065, USA; ryanc8@mskcc.org; 8Desai Sethi Urology Institute and Sylvester Comprehensive Cancer Center, University of Miami Miller School of Medicine, Miami, FL 33136, USA; nimasharifi@miami.edu

**Keywords:** prostate cancer, *HSD3B1*, genetics, genomics, transcriptomics, immunology

## Abstract

Germline variants in the *HSD3B1* gene influence response to hormone therapies. This study examined the “adrenal-permissive” allele (c.1100 C) and its impact on various somatic features in prostate tumors. Upon inferring *HSD3B1* variant status from somatic DNA sequencing data in a cohort of 6550 primary and metastatic prostate tumors, we found that tumors harboring the adrenal-permissive CC genotype, as compared to the adrenal-restrictive AA genotype, had limited somatic changes in their exomes or transcriptomes. However, significant differences were observed when we examined immune regulatory pathways and immune cell populations. This was accompanied by differential gene expression of immunoglobulins, MHC class I and II molecules, and druggable cell surface genes. These features appeared to deviate across different metastatic biopsy sites. Altogether, while tumor-intrinsic features are not regulated by germline *HSD3B1* c.1100 alleles, these genotypes instead may impact cell–cell interactions and the tumor microenvironment.

## 1. Introduction

In endocrine-driven cancers, such as prostate cancer (PC), steroid metabolism is a key driver of oncogenic progression. 3β-hydroxysteroid dehydrogenase type 1 (3βHSD1) is an enzyme encoded by a germline-regulated gene, *HSD3B1*, which plays a substantial role in steroid metabolism [1]. 3βHSD1 catalyzes the conversion of cholesterol-derived precursors into estrogens and androgens and is a critical enzyme involved in the conversion of dehydroepiandrosterone (DHEA) to testosterone and dihydrotestosterone (DHT) [1]. Due to this role in metabolism of adrenal steroids, 3βHSD1 has gained significant attention in oncology, particularly in the study of hormone-regulated cancers such as breast, endometrial, and prostate cancers [2,3]. Of particular interest is the germline missense variant of *HSD3B1* (c.1100 A>C, formerly denoted c.1245A>C, rs1047303, p.Thr367Asn, 23% allele frequency) which imparts resistance to ubiquitination and subsequent downstream degradation of the coded enzyme [1,4]. This variant results in an increase in androgen production and is consequently referred to as the adrenal-permissive allele, which exists both in heterozygous (AC) and homozygous (CC) genotype states [3]. In contrast, the adrenal-restrictive allele (AA) limits the generation of extragonadal androgens.

*HSD3B1* and its variants have previously been implicated in multiple forms of cancer. The adrenal-permissive genotypes of *HSD3B1* have been associated with estrogen-driven breast cancer via androgen aromatization and have been implicated with an increased incidence of metastatic recurrence and mortality in these patients [5]. Conversely, in endometrial cancers, the adrenal-permissive genotype is associated with better overall survival (OS), with the adrenal-restrictive genotype being associated with higher-grade endometrial tumors [2]. Potentially of the greatest intrigue is the dichotomous roles of *HSD3B1* variants in PC pathogenesis. Currently, androgen deprivation therapy (ADT), along with newer second-generation androgen receptor (AR) antagonists are the standard of care for patients with advanced or recurrent prostate cancer [6]. When patients are treated with ADT, adrenal precursors are the main source of androgens driving PC progression. Here, the permissive 3βHSD1 variant enhances the source of non-gonadal testosterone. The adrenal-permissive *HSD3B1* genotype has been associated with more aggressive and rapidly progressing PC during ADT, resulting in a poorer prognosis [3]. Furthermore, these functions of the permissive variant have thus been associated with the development of metastatic castration-resistant prostate cancer (mCRPC), worse OS, and greater mortality in patients with metastatic disease [7,8]. Given the established knowledge of 3βHSD1 in prostate cancer, the interrogation of the adrenal-permissive allele and its function has been almost exclusively examined based on measurements of germline status. In one study, the effects of the permissive variant on somatic tumoral features were evaluated in ~100 PC tumors [9]. In that prior study, the germline variants had limited associations with somatic driver genes, but tumors harboring this variant were associated with enrichment of several oncogenic signaling pathways including those that promote the cell cycle.

To build upon past work, we conducted an in silico analysis of 6500 tumor samples from both primary and metastatic prostate tumors using the Caris Life Sciences database. To our knowledge, this represents the largest sample size in a study of *HSD3B1* in PC. Distinct from other studies, we were able to determine the impact of these genetic variants based on paired whole-exome and whole-transcriptome sequencing of tumor tissue. We utilized these resources to investigate how germline *HSD3B1* variant status impacts the somatic features, signaling pathways, and immune microenvironment in prostate cancers with the goal of further elucidating how *HSD3B1* variant status may impact therapy response.

## 2. Materials and Methods

### 2.1. Patient Cohort and HSD3B1 Genotype Assignment

This retrospective study was conducted under Caris Life Sciences’ (Phoenix, AZ, USA) Research Data Banking protocol, which was reviewed and granted IRB exemption by the WCG IRB. The study adhered to the ethical guidelines of the Declaration of Helsinki, the Belmont Report, and the US Common Rule. We included all patients with submitted diagnoses of prostate cancer (n = 6500 PC) whose samples had undergone NextGen sequencing of DNA (592-gene/whole exome) and RNA (whole transcriptome) at Caris Life Sciences. The tumors were then segmented based on variant allele frequency, and genotype status was assigned. Patient-reported demographic data were compiled.

The germline *HSD3B1* c.1100 allele status was inferred from somatic DNA sequencing data, as matched tumor/germline sequencing data were not available. To do this, we evaluated the variant allele frequency (VAF) status of *HSD3B1* at position c.1100. Tumors with 0% VAF for c.1100A were considered the CC genotype, tumors with a VAF ranging from 40 to 60% for c.1100A were considered the heterozygous genotype (AC), and those with 100% VAF for c.1100A were classified as the AA genotype (Figure 1A). All other VAF ranges for c.1100A were excluded, and denoted as “uncategorized”, because we could not be confident of the germline *HSD3B1* status in those samples.

### 2.2. Whole-Exome Sequencing and Next-Generation Sequencing

NGS was performed on genomic DNA isolated from formalin-fixed paraffin-embedded (FFPE) tumor samples using the NextSeq or NovaSeq 6000 platforms (Illumina, Inc., San Diego, CA, USA). Clinically relevant genes (either 592 or 700 genes) at a high average sequencing depth of >500 with high sequencing coverage. Genetic variants identified were interpreted by board-certified molecular geneticists and categorized as ‘pathogenic’, ‘likely pathogenic’, ‘variant of unknown significance’, ‘likely benign’, or ‘benign’, according to the American College of Medical Genetics and Genomics (ACMG) standards. Copy number alterations (CNAs) were numerically determined by calculating the average depth of the sample along with the sequencing depth of each exon, which was then compared to the pre-calibrated value.

### 2.3. Whole-Transcriptome Sequencing

RNA was collected from the FPPE specimens. The Illumina NovaSeq 6500 platform was used to sequence the entire transcriptome from patients at an average of 60M reads. Raw data were demultiplexed by an Illumina Dragon BioIT accelerator, trimmed, counted, PCR duplicates were removed, and the data were aligned to the hg19 human reference genome via STAR aligner. For transcription counting, transcripts per million values were generated using the Salmon expression pipeline.

### 2.4. Immune Signatures

Immune cell signatures were inferred using deconvolution of WTS data using the quanTIseq (RRID:SCR_022993) method [10]. This included CD4/8 T cells, natural killer (NK) cells, B cells, and macrophage fractions.

### 2.5. Statistical Analysis

Kruskal–Wallis, chi-square, and Fisher’s exact tests were used to determine significance between groups. Adjusted *p*-values (*q*-value; Benjamini–Hochberg procedure) < 0.05 were considered statistically significant. A separate gene set enrichment analysis [11] (GSEA, RRID:SCR_003199, Version: v4.4.0) ranked analysis test was performed based on the log2 fold changes in CC/AA, representing differential gene expression across each specified site. The GSEA was performed with the GSEApy9 package. Normalized enrichment scores (NESs) and false discovery rates (FDRs) were used to consider significance.

### 2.6. Tumor Mutation Burden and MSI-H/dMMR

TMB was calculated by quantifying all non-synonymous mutations, including missense, nonsense, in-frame insertions or deletions, and frameshift alterations. Only somatic mutations were included in this count, excluding any variants previously characterized as germline in the dbSNP151 or Genome Aggregation Database (gnomAD). Additionally, variants deemed benign by Caris geneticists—based on criteria established by the ACMG—were also excluded from the analysis. A cutoff point of ≥10 mutations per MB was used. A combination of IHC (MLH1, M1 antibody [RRID:AB_2336022]; MSH2, G2191129 antibody [RRID:AB_2936886]; MSH6, 44 antibody [RRID:AB_2336020]; and PMS2, EPR3947 antbody [RRID:AB_2336010]; Ventana Medical Systems, Inc., Tucson, AZ, USA) and NGS was used to examine >2800 target microsatellite loci and compare them to the reference genome hg19. The two platforms generated highly concordant results as previously reported [12], and in the rare cases of discordant results, the MSI/MMR status of the tumor was determined by IHC.

## 3. Results

### 3.1. Prevalence of HSD3B1 Variants Based on Race, Treatment Status, and Tumor Biopsy Location

Using 6,550 primary and metastatic PC tumors from the Caris Life Sciences database, we first defined the prevalence of the three germline *HSD3B1* c.1100 genotypes in these patients. In this cohort, the overall prevalence of the c.1100 AA genotype was 48.8%, c.1100 AC was 32.8%, and c.1100 CC was 14.9% (Figure 1B). We further evaluated the *HSD3B1* c.1100 genotypes across self-reported race categories. The AA genotype was most prevalent across all races, followed by AC and then CC. Given the poor prognostics associated with the adrenal-permissive genotype, we further analyzed the percent of each patient population with the CC (adrenal-permissive–homozygous) allele. Prior studies have indicated that in a non-cancerous general population, the CC genotype frequencies are at 10%, 1%, and 1% for Whites, African Americans, and Asian Americans, respectively [13]. In this population of individuals that have PC, we saw general enrichment of the CC allele. White patients had the highest rate of CC at 14.3%. African American and Asian patients had lower rates of 9.5% and 10.3%, respectively (Figure 1C). The frequency of these genotypes was generally consistent among patients, who have metastatic tumors in the bone, lymph node, liver, or lung (Appendix A). Interestingly, in lung metastases, the CC genotype exhibited a prevalence of ~20%, whereas in lymph node metastases, the prevalence of CC was ~11% (Figure 1D). This suggests that the occurrence of lung metastases may be slightly greater in prostate cancer patients harboring the *HSD3B1* c.1100 CC genotype.

### 3.2. HSD3B1 C.1100 Genotypes Lack Association with AR and AR-Related Genes

Since 3βHSD1 contributes to extragonadal androgen synthesis and this activates AR signaling, we surmised that these tumors may also be enriched with other somatic features associated with enhanced AR signaling. To test this hypothesis, we analyzed genomic alterations in the AR pathway across *HSD3B1* variants. Among different *HSD3B1* c.1100 genotypes, we did not find differences in the prevalence of *AR*, *FOXA1*, or *SPOP* alterations, *AR-V7* expression, or *TMPRSS2-ERG* fusions across *HSD3B1* genotypes in both primary and metastatic tumors (Figure 2A,B). Further, among all 414 genes in which pathogenic alterations were detected, there were no significant differences between *HSD3B1* genotypes. When examining mRNA expression of AR regulatory genes (*AR*, *FOXA1*, and *HOXB13*) across the three *HSD3B1* genotypes, we also did not find significant differences, as we had initially anticipated (Figure 2C). Finally, we sought to compare the expression of AR and neuroendocrine gene signatures [14] among the *HSD3B1* genotypes. Consistent with our previous finding, we noted that none of the genotypes were significantly associated with respect to AR or neuroendocrine signatures (Figure 2D). Altogether, when examining tumoral features associated with PC, we found minimal somatic alterations that were associated with the c.1100 AA or CC *HSD3B1* genotypes.

### 3.3. HSD3B1 Variants Have Limited Impact on the Transcriptome but Regulate Hallmark Signaling Pathways

Given the limited influence on genomic alterations, we hypothesized that differences between the c.1100 AA and CC *HSD3B1* genotypes could be detected in the tumor transcriptomes. We thus analyzed the transcriptional differences of >40,000 genes detected by the Caris Life sciences platform in these PC tumors (Appendix A). In primary PC tumors, we found no significant differences in the differentially expressed genes across these two homozygous *HSD3B1* genotypes (c.1100 AA versus CC). In metastatic PC tumors, two genes were both significantly upregulated in bone and lung metastases in the AA genotype: *SEBOX*, which plays a role in embryonic development [15], and *ZNF177*, which aids in maintaining genomic stability [16] (Figure 3A). Despite the overall limited number of differentially expressed genes, we observed significant differences in oncogenic and immune pathway signatures when directly comparing all specimens that harbored the CC genotype relative to the AA genotype. Using gene set enrichment analysis (GSEA) [17], focusing on primary PC tumors and lung metastases, the CC genotype conferred almost exclusively downregulated hallmark oncogenic and immune pathway signatures compared to tumors with the AA genotype (Figure 3B,C). Conversely, in bone metastases, the CC genotype almost exclusively conferred trends of increased hallmark immune and oncogenic pathways. When focusing on immune pathways, metastatic tumors to the bone, lymph node, and liver generally exhibited trends towards enhanced immune pathways when directly comparing the CC to the AA genotype. Interestingly, in prostate samples and liver metastases, the CC genotype conferred reduced hallmark androgen response. Altogether, these differences indicate that the *HSD3B1* c.1100 genotypes, albeit exhibiting limited change in individual genes, had a significant impact on signaling pathways and immune regulatory pathways in a tumor-site-specific manner.

### 3.4. HSD3B1 Genotypes Are Associated with Differences in the Immune Microenvironment

As androgens and their signaling pathways may play a role in modifying immune cell features [18,19], we analyzed immune cell infiltration across *HSD3B1* c.1100 variants in the same PC cohort. Using quanTIseq [10], we assessed levels of immune cells including B cells, natural killer cells, CD4/8 T cells, and regulatory T cells (Appendix A). While statistical significance was not observed, selected trends recapitulated our findings in the hallmark immune-related signatures (Figure 3C). Particularly, in lung metastases, we observed that the CC genotype conferred lower median levels of CD4/8 and regulatory T cells (Figure 4A). In liver metastases, we saw an increase in CD4/8 and regulatory T cells in the CC relative to the AA genotype. Several trends were observed in other metastatic sites but were not consistent across the three types of T cells. Finally, we quantified the prevalence of markers of immune checkpoint inhibitors (ICIs) for PC [20], including tumor mutational burden (TMB) and microsatellite instability (MSI). From this, we did not find that lung metastases with the CC genotype were either TMB-high or MSI-high. Rather, the lung metastases that were TMB-high or MSI-high exclusively harbored the AA or AC genotypes (Figure 4B). Liver metastases that were of the CC genotype had increased TMB-high and MSI-high rates as compared to the AA genotype. Altogether, when directly comparing immune cell fractions or markers of ICIs among the *HSD3B1* c.1100 CC and AA variants, it was apparent that the adrenal-permissive allele (CC) was associated with significant immune microenvironment differences in lung and liver metastases.

### 3.5. HSD3B1 Variants Are Associated with Divergent Expression of Immune Regulatory Features and Cell Surface Genes in a Tissue-Specific Manner

In a unique case study, we previously identified a PC patient with two tumors that exhibited a grossly different pattern of immune cell interactions [21], which was accompanied by divergent expression patterns of class I/II MHC genes and immunoglobulin genes, as well as a suite of cell surface proteins. Of these, some were immune checkpoint molecules (PD-1, PDL1, and CTLA4), whereas others were current experimental targets for antibody drug conjugates, CAR-T cells, or bi-specific T-cell engagers (DLL3, STEAP1, and PSMA) [22,23,24]. Accordingly, we sought to explore these features according to the different *HSD3B1* c.1100 genotypes. Similar to the GSEA and immune cell profiling, we found that whether we examined MHC class I/II genes (Figure 5A), immunoglobulin genes (Figure 5B), or cell surface targets (Figure 5C), the trends indicated differences in relative expression when directly comparing levels in CC to AA homozygous genotypes. Importantly, like the observations in signaling pathways and immune cell repertoires, it was clear that these differences were site-specific. In lung metastases, the CC genotype conferred reductions in almost all MHC class I/II and immunoglobulin genes, as well as several cell surface target genes. While the expression levels appeared stochastic across the other sites, there were notable trends that reflect the GSEA analyses, as primary prostate specimens generally had lower immunoglobulins, whereas subsets of MHC class I/II molecules and immunoglobulins were increased in liver metastases. Altogether, when directly comparing the CC and AA homozygous genotypes in PC with consideration of tissue site, there are significant differences in genes that regulate immune cell interactions and cell surface molecules for targeted therapies.

## 4. Discussion

Prior studies largely focused on the impact of the *HSD3B1* genotypes and their association with responses to ADT or outcomes in PC patients [4,7,8]. Using samples from the Caris database, we interrogated the association between *HSD3B1* germline variants and somatic features in diseased prostate tumor tissue. To our knowledge, this is the largest study that has examined the germline and somatic interactions of PC patients based on the *HSD3B1* c.1100 genotypes. While tumors harboring the CC genotypes were expected to have increased dependency on AR signaling, this was not supported by our analysis of *AR* alterations, expression of AR-related genes, or expression of AR signaling activity. Our study instead depicts an association of the *HSD3B1* variant status as a potential regulator of the tumor microenvironment. This notion was supported by differences in immune-associated hallmark pathways and immune cell fractions, as well as the expression of immunoglobulins, MHC class I/II molecules, and cell surface targets. Importantly, it was clear that these observations were highly dependent on metastatic site, and these effects were most notable in lung metastases that were also enriched for the CC (adrenal-permissive–homozygous) genotype.

The *HSD3B1* adrenal-permissive genotype yields a stable protein that enhances the production of high-affinity ligands to the androgen receptor [1]. As the rate-limiting step to androgen production, some inhibitors against the 3βHSD1 protein product for prostate cancer have been developed, and it has been found that the *HSD3B1* CC genotype predicts worse outcomes in PC patients that receive hormone therapies [4,7,8]. By capturing the frequency of allele status across races in the setting of PC and potentially mCRPC, we noted there was a general enrichment of the CC genotype compared to a non-cancerous general population. In White, African American, and Asian populations, the CC genotype frequencies were reported to be 10%, 1%, and 1%, respectively [13]. In our PC cohort, we saw enrichment of the CC genotypes for White, African American, and Asian patients with rates of the CC genotype of 14.3%, 9.5%, and 10.3%, respectively. After all, in the general population, few men develop later stages of PC. This supports the relevance of CC genotypes in the development and progression of PC. However, of the many genomic and transcriptomic alterations examined, we found limited evidence that tumoral AR signaling was increased in patients harboring the CC genotype. Further, based on genomic and transcriptomic analyses, there were limited somatic features differentially associated with the CC or AA variants. While this may be surprising, this is concordant with a prior study conducted on 101 PC samples in which their whole genome was sequenced [9]. In that study, the adrenal-permissive genotypes exhibited limited association with pathogenic genomic alterations typically observed in PC. While the transcriptomic alterations were generally limited, in bone and lung metastatic tumors, the AA genotype was enriched with *SEBOX* [15] and *ZNF177* [16] expression. While these genes are currently not implicated in prostate cancers, future studies are required to determine their relevance in prostate cancer progression.

Our novel findings indicated that the adrenal-permissive state of *HSD3B1* is associated with differences in the immune microenvironment of prostate tumors, but in a tissue-site-specific manner. We specifically observed that tumors with *HSD3B1* c.1100 CC and AA genotypes had notable differences in hallmark immune regulatory pathways, immune cell fractions, immunoglobulins, and MHC class I/II genes. While many of these differences were not statistically significant, we argue that the concordance of these changes was notable. Particularly, in ~20% of lung metastatic samples, the CC genotype was immune-suppressive, with concordant decreases in B cells, natural killer cells, CD4/8 T cells, T-regulatory cells, and essentially all MHC class I/II and immunoglobulin genes in the groups of patients. While we did not find increases in tumoral AR signaling in this study, androgens are known to modulate various features of immune cells and response to immune therapies [25,26], perhaps due to their potential role in regulating T-cell exhaustion [27,28]. This coincides with other findings that indicate that blockade of AR activity enhances the efficacy of T-cell therapies [18,19]. In this setting, Chesner et al. recently showed that ablation of androgen signaling regulates immune therapy response by a process that increases MHC class I expression [18]. In our study, there were no measurements of local androgen levels; therefore, it remains to be determined if the CC genotype in fact enhances AR activity in the peripheral immune cells through elevated production of local androgens, or if the CC genotype somehow influences the immune cells. While these are compelling mechanisms, we also note that in lung metastases and primary prostate biopsies, the CC genotype was associated with increased hallmark immune pathways and expression of immunoglobulins and MHC class I/II genes. In liver metastases, tumors harboring the CC genotypes had numerically increased mean levels of CD4/8 and regulatory T cells. Altogether, it is worth noting that *HSD3B1* genotypes, regulated by local tissue type, may impact the tumor microenvironment, and that this process is also regulated by local tissue type. This differential expression of genes expressed on the prostate tumor cell surface, or genes that are immune therapy targets, indicates a need to evaluate *HSD3B1* genotypes when considering these therapies. Further, since findings were specific to metastasis sites, these are factors that must be considered.

The *HSD3B1*-permissive allele is also thought to regulate the conversion of steroid precursors to estradiol [5], the high-affinity ligand for estrogen receptors (ERs). Like AR in PC, ERs drive a large subset of breast cancers that are treated with hormone therapies. Studies have indicated that the CC genotype is significantly enriched as compared to the AA genotype in ER+ breast cancers [5]. Further, the CC genotype is associated with poor outcomes in ER+ patients [5], particularly in post-menopausal women with reduced hormone levels. Subsets of endometrial tumors are also thought to be driven by ER activity [29,30]. Interestingly, our prior study of samples from the Caris database indicated that in post-menopausal women, endometrial tumors harboring the CC genotype were enriched with multiple genomic alterations in the PI3 kinase pathway [31], including *PTEN*, *PIK3CA,* and *PIK3R1*. This was not observed in pre-menopausal women. Altogether, while the observations in PC indicate that variant status of *HSD3B1* has limited impact on the tumor-intrinsic genome, *HSD3B1* variant status may impact the tumor genome in other cancers and different clinical settings. These findings suggest that the function of *HSD3B1* may complement other oncogenic signaling pathways, such as estrogen or PI3 kinase, in breast or endometrial cancers.

### Limitations of the Study

Our study had several inherent limitations, including its retrospective nature. We also lack granular clinical information, including grade, disease stage, or prior treatments. The immune deconvolution of these tumor samples was inferred from bulk RNA sequencing rather than sequencing of single immune cells. Finally, we organized the data based on self-reported race, whereby certain patients were uncharacterized and excluded from analysis.

## 5. Conclusions

This work uniquely examines the impact of inferred germline *HSD3B1* c.1100 variants on somatic features in a large cohort of primary and metastatic PC tumors. Our study indicates that in prostate tumors, *HSD3B1* germline status is associated with novel functions beyond androgen signaling. While the CC and AA genotypes do not directly influence somatic genomic landscapes, they lead to pronounced differences in immune cell repertoires, immune regulatory genes and pathways, and cell surface target expression. These differences were diverse when considering metastatic sites, especially in lung metastases that are enriched for the CC (permissive–homozygous) genotype. While our conclusions must be substantiated in further research, the function of *HSD3B1* genotypes extends beyond responses to androgen-directed therapies and should be considered when using alternative therapies that target immune cells or those that engage cell surface drug targets.

## Figures and Tables

**Figure 1 cancers-17-01270-f001:**
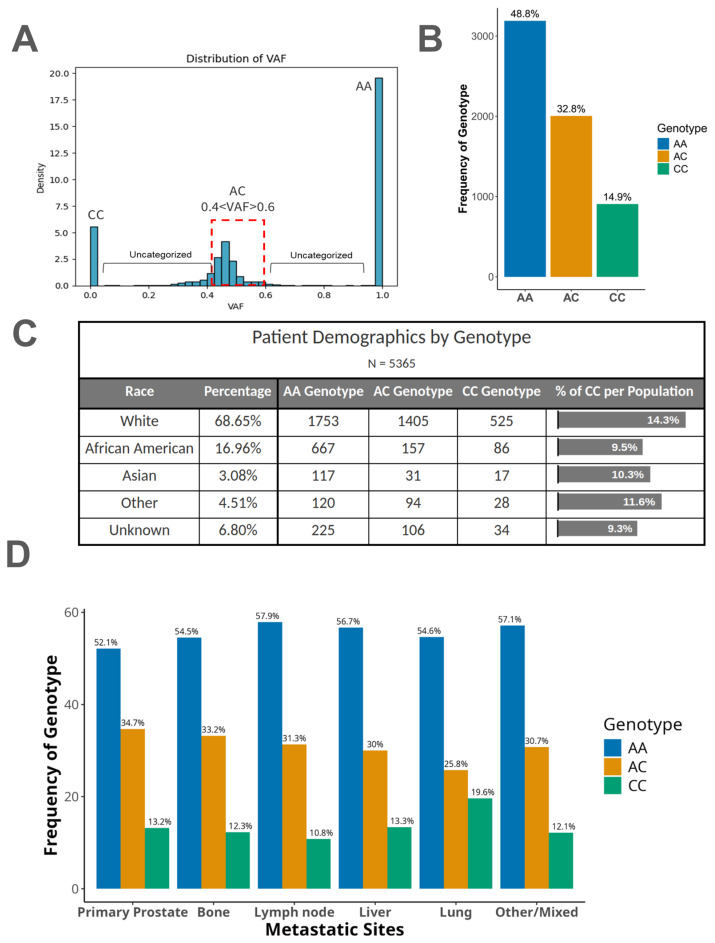
(**A**) Distribution of variant allele frequencies (VAFs) for *HSD3B1* c.1100A. Tumors that had 100% A allele detection were considered AA homozygous, 0% c.1100A allele detection were considered CC homozygous, and from 40 to 60% c.1100A allele detection were considered heterozygous. (**B**) Distribution of genotypes based on *HSD3B1* variant c.1100 allele classification based on the VAF analysis for the entire cohort of PCs. (**C**) Distribution of VAF-determined genotypes stratified by self-reported patient race. (**D**) Distribution of *HSD3B1* c.1100 VAF-assigned genotypes across PC metastatic sites.

**Figure 2 cancers-17-01270-f002:**
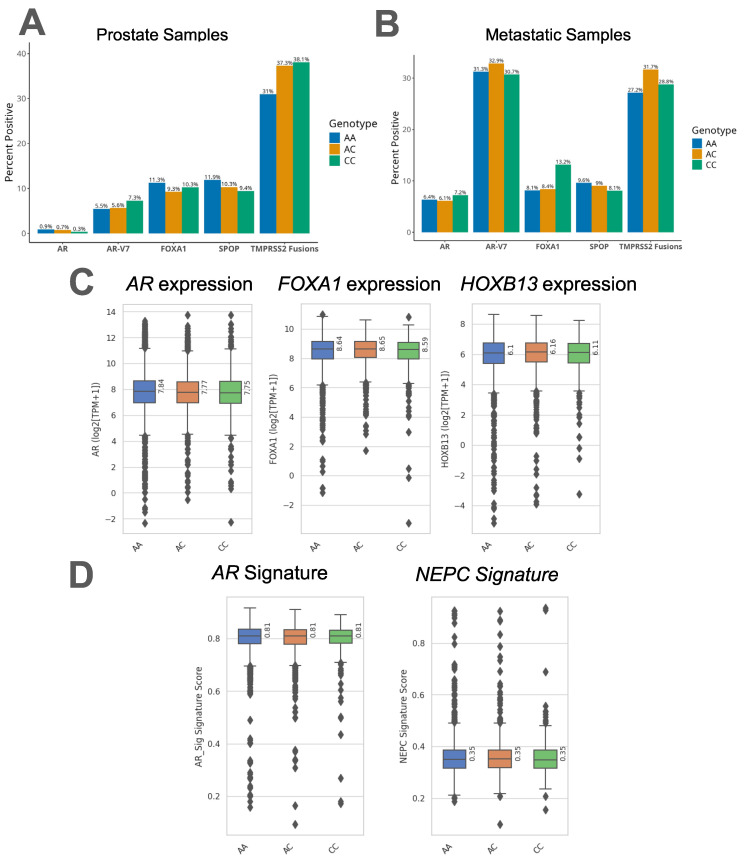
The prevalence of genes in the AR pathway and their rates of alteration are shown by *HSD3B1* genotypes in (**A**) primary prostate and (**B**) metastatic biopsies. Organized by *HSD3B1* genotype, we depict (**C**) mRNA expression levels of AR regulatory genes are shown for *AR*, *FOXA1*, and *HOXB13*. (**D**) AR and NEPC signature scores are shown based on each *HSD3B1* genotype.

**Figure 3 cancers-17-01270-f003:**
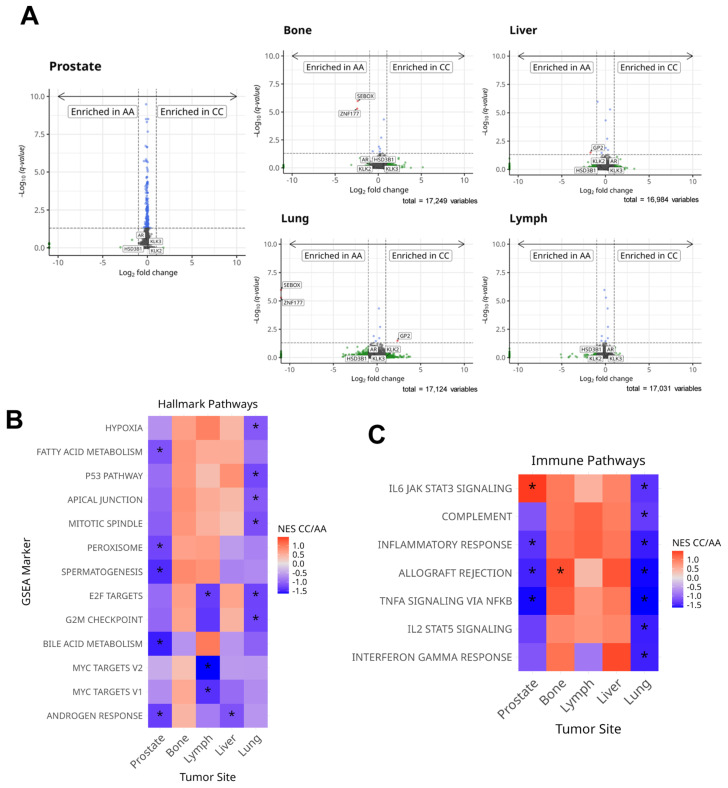
(**A**) Volcano plots depict the log2 fold change in genes across *HSD3B1* c.1100 CC and AA genotypes organized by tumor biopsy site; *AR*, *KLK2*, *KLK3*, and *HSD3B1* are highlighted. GSEA was deployed to determine the differences in hallmark oncogenic (**B**) and immune (**C**) pathways. * FDR < 0.05.

**Figure 4 cancers-17-01270-f004:**
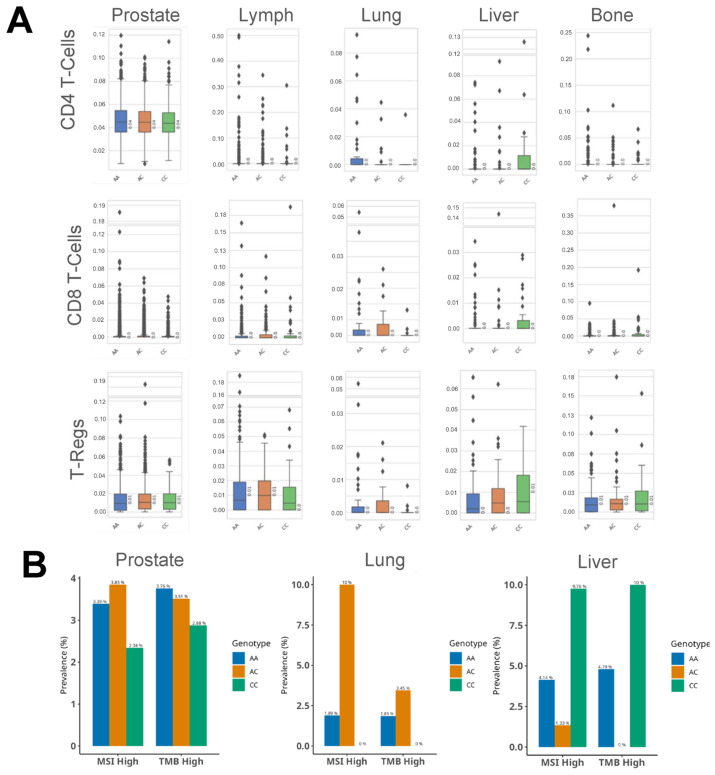
(**A**) QuanTIseq-inferred T-cell infiltration based on *HSD3B1* c.1100 genotypes stratified by tumor site. (**B**) Percentage of TMB-high and MSI-high samples in tumors from the prostate, lung, and liver, stratified by the three *HSD3B1* genotypes.

**Figure 5 cancers-17-01270-f005:**
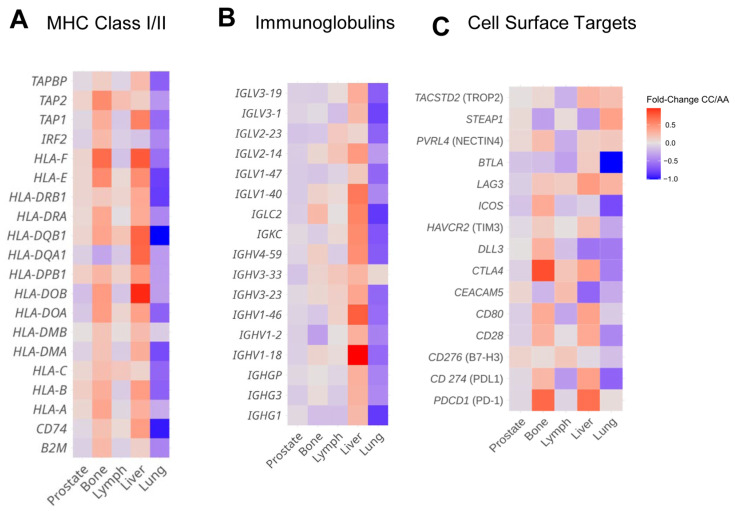
Comparisons of *HSD3B1* c.1100 CC and AA genotypes. Log2 fold change levels are depicted for transcripts based on groups of genes that are (**A**) MHC class I/II molecules, (**B**) immunoglobulins, and (**C**) cell surface targets for targeted therapies in PC.

## Data Availability

This study did not generate any new raw DNA/RNA sequencing data but rather used existing data from the Caris Life Sciences POA database through a formal letter of intent and subsequent data use agreement. Here, we researched the de-identified data collected in a real-world health care setting and this is subject to controlled access for privacy and proprietary reasons. When possible, derived data supporting the findings of this study have been made available within the paper and its Appendix A. Other data can be acquired through a letter of intent to Caris Life Sciences (https://www.carislifesciences.com/letter-of-intent/). Additional inquiries can be sent to Andrew Elliott at aelliott@carisls.com.

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
