# Peer review of "The Influence of the Germline HSD3B1 Adrenal-Permissive Allele (c.1100 C) on the Somatic Alteration Landscape, the Transcriptome, and Immune Cell Infiltration in Prostate Cancer"

_cancers, 2025, doi:10.3390/cancers17081270_

Round 1

Reviewer 1 Report

Comments and Suggestions for Authors

Dear Authors,

The article is good overall but requires minor revisions before being considered for publication. Kindly see the points below for the details:

Line 92: The typo in "paticularlly" should be "particularly.”

Line 174: Restructure this sentence for better elaboration [benign variants identified by Caris geneticists….]

Line 210-213 (Figure 1 legend): Could you briefly describe how genotypes were classified to improve clarity?

Line 313-315: The term [cell surface targets for targeted therapies in PC] is vague, so please consider specifying a few key targets for clarity.

Line 316-326: The discussion is well-structured, but a sentence summarizing key implications for future prostate cancer treatment would strengthen the conclusion.

Line 450: Ensure at least a few new recent (2023-2025) references are included to reflect the latest advancements.

Reviewer 2 Report

Comments and Suggestions for Authors

I have no comments, I really appreciate the work and strongly recommend the manuscript to the next step. I have seen only few typos issues which authors will address it.

Reviewer 3 Report

Comments and Suggestions for Authors
  1. Authors need to provide the RRID of all the resources used in this study.
  2. In the Results 3.1 section, from line 201 to 207, authors are encouraged to rephrase the sentences. As the HSD3B1 c.1100 genotypes (AA and CC only) are considered as germline, “….frequency of these genotypes was generally consistent across common metastatic sites such as bone, lymph node, liver, and lung. ….” indicates the HSD3B1 c.1100 as somatic variants. It can be rewritten as ‘….frequency of these genotypes were generally consistent among patients, who have the metastatic sites of bone, lymph node, liver,….’ and so on and so forth.
  3. Authors must provide the HSD3B1 c.1100 variant details as the SNP ID, the protein product, and genomeAD minor allele frequency.
  4. Authors are encouraged to consolidate the statements about the limitations of this study in the main text in a separate heading ‘Limitations of the study’.
